# Repurposing Drugs in Small Animal Oncology

**DOI:** 10.3390/ani13010139

**Published:** 2022-12-29

**Authors:** Antonio Giuliano, Rodrigo S. Horta, Rafael A. M. Vieira, Kelly R. Hume, Jane Dobson

**Affiliations:** 1CityU Veterinary Medical Centre, City University of Hong Kong, Kowloon, Hong Kong; 2Department of Veterinary Clinical Sciences, Jockey Club College of Veterinary Medicine, City University of Hong Kong, Kowloon, Hong Kong; 3Department of Veterinary Medicine and Surgery, Veterinary School, Universidade Federal de Minas Gerais, Belo Horizonte 31270-901, MG, Brazil; 4Clínica Veterinária Saúde Única, São Bernardo do Campo 09726-150, SP, Brazil; 5Department of Clinical Sciences, College of Veterinary Medicine, Cornell University, Ithaca, NY 14850, USA; 6Department of Veterinary Medicine, University of Cambridge, Cambridge CB3 0ES, UK

**Keywords:** repurposing drugs, oncology, small animal oncology

## Abstract

**Simple Summary:**

Repurposing drugs in oncology consists of using off-label drugs that are licensed for various non-oncological medical conditions to treat cancer. Repurposing drugs has the advantage of using drugs that are already commercialized, with known mechanisms of action, proven safety profiles, and known toxicology, pharmacokinetics and pharmacodynamics, and posology. In this review, we summarize both the benefits and challenges of repurposing anti-cancer drugs; we report and discuss the most relevant studies that have been previously published in small animal oncology, and we suggest potential drugs that could be clinically investigated for anti-cancer treatment in dogs and cats.

**Abstract:**

Repurposing drugs in oncology consists of using off-label drugs that are licensed for various non-oncological medical conditions to treat cancer. Repurposing drugs has the advantage of using drugs that are already commercialized, with known mechanisms of action, proven safety profiles, and known toxicology, pharmacokinetics and pharmacodynamics, and posology. These drugs are usually cheaper than new anti-cancer drugs and thus more affordable, even in low-income countries. The interest in repurposed anti-cancer drugs has led to numerous in vivo and in vitro studies, with some promising results. Some randomized clinical trials have also been performed in humans, with certain drugs showing some degree of clinical efficacy, but the true clinical benefit for most of these drugs remains unknown. Repurposing drugs in veterinary oncology is a very new concept and only a few studies have been published so far. In this review, we summarize both the benefits and challenges of using repurposed anti-cancer drugs; we report and discuss the most relevant studies that have been previously published in small animal oncology, and we suggest potential drugs that could be clinically investigated for anti-cancer treatment in dogs and cats.

## 1. Introduction

In the past few years, the repurposing of drugs in oncology has become a common topic of discussion. Repurposing drugs in oncology consists of using off-label drugs that are licensed for various non-oncological medical conditions to treat cancer [1]. The oldest drugs that have been repurposed for cancer treatment are non-steroidal anti-inflammatory drugs (NSAIDs). The benefits of aspirin for the prevention and treatment of colon cancer in humans is widely known [2,3]. Similarly, in veterinary medicine, the use of piroxicam has proven effective in the treatment of urothelial carcinoma in dogs [4]. Other successful examples in humans include the response to clarithromycin in human patients with chronic myeloid leukemia, retinoids in acute promyelocytic leukemia, propranolol in angiosarcoma, and thalidomide in multiple myeloma [5,6,7,8,9,10]. In recent years, numerous other drugs used to treat various conditions have been claimed to have possible “anti-cancer effects”. Most claims arise from epidemiology studies that have found a low incidence of cancer or improved survival in people receiving such drugs [11,12,13]. A classic example is the cancer survival benefit, found in type 2 diabetic patients treated with metformin [13,14]. The interest in repurposed anti-cancer drugs has led to a few randomized clinical trials in humans, with certain drugs showing some degree of clinical efficacy [15]. However, despite the enthusiasm for the repurposing drug strategy, only a few of them have been approved for the treatment of cancers [16].

## 2. The Advantage of Repurposing Drugs in Oncology Is Very Clear—Low Cost, Low Side Effects, Easy Access, and Worldwide Availability [16]

The approval of a new anti-cancer drug involves a series of steps, from the discovery, isolation, and chemical optimization of a new compound, to lengthy preclinical efficacy and safety trials *in vitro* and *in vivo*, to costly phase I, II, and III human clinical trials, before the final regulatory approval. These processes, although essential, are lengthy and require large investments from pharmaceutical companies. Hence, it is not surprising that newly discovered anti-cancer drugs are very expensive. As a result, large numbers of people in many low-income countries do not have access to most of the new anti-cancer medications [17,18]. Repurposing drugs has the advantage of using drugs that are already commercialized, with known mechanisms of action, proven safety profiles, known toxicology, pharmacokinetics, and pharmacodynamics, and posology. These drugs are usually much cheaper than new anti-cancer drugs and, even in low-income countries, they are widely affordable, especially when a generic compound with multiple manufacturers is available [16]. In veterinary oncology, the consideration of drugs’ costs and safety profiles is even more important than in human oncology, and drug repurposing could be an appealing strategy to treat pets with cancer. However, excluding the popular COX-2 inhibitors (piroxicam/meloxicam/firocoxib, etc.), only a few studies have been published on the use of repurposed drugs in veterinary oncology. There are numerous drugs that are potentially promising candidates to be repurposed for use in veterinary oncology. The purpose of this review is to discuss the main drugs that, in the authors’ opinion, could be potentially cost-effective and appropriate for further investigation in pets. Side effects of repurposed drugs are not a focus of this review, since these drugs are already in common use for other conditions, and thus will not be discussed in detail in this review. 

## 3. Repurposed Drugs Investigated in Clinical Setting in Dogs and Cats

### 3.1. Auranofin

Auranofin is a drug used to treat rheumatoid arthritis in humans, with a newly discovered anti-cancer effect in vitro. It has recently gained attention as a possible treatment for osteosarcoma in dogs [19,20,21]. Auranofin’s main anti-cancer mechanism is the inhibition of thioredoxin reductase, an enzyme responsible for maintaining the redox balance via reducing the excessive intracellular buildup of detrimental reactive oxygen species (ROS). Auranofin induces excessive levels of intracellular oxidative stress, leading to cancer cell death [22,23]. A single-arm phase I/II multicenter study investigated the safety and efficacy of auranofin combined with a standard treatment (amputation followed by carboplatin chemotherapy) for dogs with osteosarcoma. The study concluded that there was a survival advantage for dogs treated with auranofin, compared to the historic control group treated only with the standard of care. Unusually, the survival advantage was significantly more pronounced for male dogs compared to females, and the underlying reason for this was not clear [20]. 

### 3.2. Desmopressin

Desmopressin (1-deamino-8-d-arginine vasopressin or DDAVP) is a synthetic derivative of an anti-diuretic hormone (vasopressin), used to treat diabetes insipidus [24,25]. DDAVP is also used for the prophylaxis of bleeding disorders prior to surgery, due to the ability to release factor VIII and von Willebrand factor [26]. The proposed anti-metastatic effect of desmopressin seems to relate to its ability to inhibit metastatic emboli formation and the subsequent adherence of emboli to target metastatic sites [27,28,29,30]. Studies in mouse models have shown the ability of desmopressin to reduce the formation of melanoma and mammary carcinoma metastasis [27,30,31]. 

Intraoperative desmopressin has been investigated for the prevention of mammary carcinoma metastasis in dogs and cats [32,33,34,35]. One small pilot study investigated the efficacy of pre-operative treatment with DDAVP in twenty-one dogs with malignant mammary carcinoma [33]. The patients were randomly allocated to group 1, consisting of 11 cases treated with intravenous DDAVP 30 min prior to and 24 h after surgical treatment, and group 2, consisting of 10 patients treated with a placebo. The study found a significant increase in the disease-free interval and survival time for dogs treated with DDAVP compared to the placebo and concluded that the administration of DDAVP could prolong the disease-free and overall survival in dogs with mammary carcinoma [33]. In a similar small study, the same authors investigated the efficacy of desmopressin in twenty-eight dogs with various grades of mammary carcinoma (18 DDVAP- and 10 placebo-treated dogs) [34]. The authors found that DDAVP significantly increased Disease Free Interval (DFI) and Median Survival Time (MST) in grade II and III mammary carcinoma [34]. A more recent prospective randomized study investigated DDAVP in twenty-four patients with canine mammary carcinoma [35]. The study found no difference in DFI and MST between DDAVP- and placebo-treated groups and concluded that perioperative desmopressin does not prevent metastasis in dogs with mammary carcinoma [35]. It is interesting to note that despite all studies using a similar DDAVP treatment protocol, the study that did not find any difference in survival used a subcutaneous injection of DDAVP, compared to the intravenous injection used in the earlier studies. A recent retrospective study investigated the benefit of intravenous preoperative DDAVP treatment in sixty cats with mammary carcinoma treated with bilateral mastectomy [32]. The study found no survival advantage for the DDAVP-treated cats; however, only 15 cases received DDAVP treatment [32].

### 3.3. Doxycycline

Doxycycline is a low-cost, broad-spectrum, oral antibiotic commonly used in veterinary medicine [36,37]. Doxycycline has shown some anti-proliferative effects in vitro and in a xenograft tumor model in mice [38,39]. Doxycycline has anti-angiogenic properties and can inhibit matrix metalloproteinase (MMPs) and nuclear factor kappa-light-chain-enhancer of activated B cells (NF-kB) signaling, which are important in the development and progression of cancer [40,41,42]

Recently, doxycycline has been found to reduce mitochondrial protein synthesis and oxidative phosphorylation capacity, decrease the uptake of glucose from neoplastic cells, and increase the extracellular acidification rate in vitro [43]. In an oral squamous cell carcinoma mouse model, doxycycline decreased the number of lymph node metastases from 80% to 20% [43]. 

In one small phase II study, thirteen dogs with multicentric B-cell lymphoma were treated with doxycycline as a sole agent for 1 to 8 weeks, and none of the patients achieved a complete or partial response, but one dog achieved stable disease for 6 weeks [44]. 

### 3.4. Losartan

Losartan is an angiotensin II type 1 receptor antagonist used to treat hypertension in humans. Telmisartan and losartan have also been used to treat hypertension and proteinuria in dogs and cats [45,46]. Losartan inhibits monocyte recruitment via inhibition of chemokine receptor and its ligand (CCL2–CCR2). Tumor and stromal cells at metastatic sites produce and release CCL2, which recruits monocytes expressing CCR2. Monocytes in early metastatic lesions differentiate into metastasis-associated macrophages, which play essential roles in metastasis formation and growth [45,46]. Losartan also has an anti-fibrotic effect via the inhibition of the TGFβ pathway [47,48]. The inhibition of fibrosis in vitro and in a mouse model improved the perfusion and delivery of chemotherapy, indicating a possible benefit of losartan in combination with standard chemotherapy treatment [47,49,50]. Indeed, in a small phase II study in humans with advanced pancreatic cancer, the addition of losartan to standard neoadjuvant chemotherapy has shown some promising results [49,51].

Losartan has been recently investigated in dogs with metastatic osteosarcoma, previously treated with the standard treatment of amputation and conventional chemotherapy [52]. In this study, twenty-eight patients with osteosarcoma lung metastases were treated with losartan at 1 mg/kg (standard antihypertensive dose in humans) and with a ten-fold increased dose of 10 mg/kg in combination with toceranib phosphate. The treatment was found to be safe, and although clinical responses were observed in both treatment groups, the higher dosage of 10 mg/kg was required for monocyte blockade. At this dose, monocyte migration and CCL2 secretion by osteosarcoma cells in vitro were significantly inhibited. Dogs treated at 10 mg/kg in combination with toceranib phosphate achieved a 50% clinical benefit with a 25% objective response [52]. Despite the lack of non-treated control patients or toceranib-only-treated patients, the authors concluded that the clinical benefit of losartan and toceranib was superior compared to previous studies of dogs treated only with toceranib phosphate [53,54,55].

### 3.5. Metformin

Metformin (1,1-dimethylbiguanide) is a semi-synthetic oral hypoglycemic agent recommended as the first-line treatment for type 2 diabetes mellitus in humans. It is an inhibitor of the mitochondrial respiratory complex I, NADH dehydrogenase, resulting in inhibition of mitochondrial oxidative phosphorylation and a reduction in ATP synthesis. The reduction in ATP production activates the adenosine monophosphate protein kinase (AMPK) signaling pathway, which ultimately results in the stimulation of glucose uptake, fatty acid oxidation in the muscles and liver, and the inhibition of hepatic glucose output and cholesterol and triglyceride synthesis [56,57]. Metformin has been recently repurposed in clinical trials for the treatment of various cancers, with increasing evidence of efficacy in the treatment and prevention of cancer in humans [58,59,60,61]. *In vitro* studies enabled the recognition of specific anti-tumor mechanisms that may occur in a concentration-dependent manner in in a variety of cancer cell lines, including pancreatic, prostatic, lung, ovarian, breast, and thyroid [58,59]. The main anti-cancer effect is related to the inhibition of the mammalian target of rapamycin (m-TOR) signaling through AMPK-dependent and -independent pathways. Activated AMPK phosphorylates the tuberous sclerosis complex protein 2 (TSC2), which inhibits m-TOR complex 1 (m-TORC1) [60]. Nevertheless, in some cells, metformin does not induce AMPK upregulation, and its activation is not necessarily translated into the inhibition of the m-TORC1 signaling [58]. Downregulation of the m-TOR pathway can still occur in such cases through alternative pathways [58,62]. Deregulation of the AMPK/AKT/m-TOR pathway is recognized as an important target in several cancers, including osteosarcoma, human breast cancer, and triple-negative feline mammary carcinomas [58,59]. In human breast carcinoma cells, metformin was able to decrease the level of epidermal growth factor 2 (HER-2), through the inhibition of protein synthesis (mediated by the m-TOR pathway) [63]. There is also evidence of the metformin-induced overexpression of p53 and decreased expression of G1 cyclins (i.e., cyclin D1), leading to cell cycle arrest [59,60]. As for the m-TOR signaling, p53 overexpression also occurs depending on or independently of AMPK activation. Further anti-tumoral effects could also be related to a decrease in serum glucose and insulin-like growth factor-1, fatty acid synthesis, and intracellular ATP [59].

A meta-analysis study showed that treatment of type 2 diabetes mellitus with metformin significantly reduced the incidence of pancreatic cancer in humans [59]. It also suggested higher overall survival for pancreatic cancer patients treated with metformin. The same study suggested an association of metformin intake with improved survival in colorectal cancer patients and a reduced risk of colorectal cancer [59].

*In vitro* studies in dogs showed that metformin administration resulted in increased apoptosis in canine prostate and bladder cancer cells [64]. Cell cycle arrest was demonstrated in canine mammary gland tumor cells [57,65]. However, while ATP depletion and increased reactive oxygen species were observed to a similar extent in metastatic and non-metastatic cell lines, AMPK activation and m-TOR inhibition occurred only in metastatic cells [57]. In the same study, metformin also suppressed tumor growth in vivo in xenografted metastatic canine mammary gland tumor cells [57]. 

In cats, injection site sarcoma cells lines were treated with metformin, leading to apoptotic or necrotic cell death, independent of m-TOR inhibition [58]. In a pilot study with nine cats (five carcinomas, two cutaneous lymphomas, and two injection site sarcomas) treated with metformin at a maximum tolerated dosage of 10 mg/kg for 12 h, only two cats, with skin carcinomas, had a modest measurable reduction in tumor size. Side effects were mostly mild to moderate and included anorexia, vomiting, and weight loss [66]. The combination of metformin with other m-TOR inhibitor drugs such as sirolimus or its combination with conventional and target chemotherapy could potentially increase its efficacy in the treatment of various cancers in dogs and cats.

### 3.6. Sirolimus

Sirolimus, also known as rapamycin (a metabolite produced by *Streptomyces hygroscopicus*), is an immunosuppressive drug originally discovered as an antifungal medication [67]. Sirolimus binds to a family of intracellular binding proteins termed FKBPs (FK binding proteins) and such complexes act as a specific inhibitor of m-TOR (Figure 1), whose pathway is involved in cell growth and metabolism, and it is deregulated in many cancer types, including osteosarcoma [67,68,69]. Sirolimus has anti-cancer effects in vitro and some efficacy in combination with chemotherapy or other anti-cancer drugs [70,71]. However, its efficacy as a single agent in patients with resistant and advanced solid tumors appears limited [72]. A large prospective and randomized clinical trial has been recently conducted in 324 dogs with osteosarcoma treated with sirolimus [73]. After completing standard treatment for osteosarcoma with amputation and carboplatin chemotherapy, dogs were non-blindly assigned and treated with oral sirolimus at 0.1 mg/kg or with no further treatment. The drug was very well tolerated and only 20% of dogs developed mild gastrointestinal side effects, but the results of this study did not indicate any significant difference in the disease-free interval and survival time between dogs treated with sirolimus and untreated patients. However, although the activation of the sirolimus target m-TOR/PIK is common in canine osteosarcoma cells, the dysregulation/activation of the drug target was not investigated in any of the groups [69,73,74]. Furthermore, pharmacokinetic modeling of sirolimus administration predicted that the target trough concentrations of 10–15 ng/mL would not be achieved, and sirolimus blood levels in most dogs were below 10 ng/mL [74].

### 3.7. Thalidomide

Thalidomide was first discovered in 1957 and was widely used as a sedative. It was later withdrawn from the market, due to significant teratogenicity when taken early during pregnancy [75]. Decades later, thalidomide was discovered to have anti-inflammatory and anti-cancer effects, mainly via decreasing tumor necrosis factor (TNF)-alpha and its anti-angiogenetic property [76]. Due to the anti-angiogenic, immune-modulatory benefit and the clinical efficacy of thalidomide in people with multiple myeloma, some studies have investigated the efficacy of this drug in dogs. In veterinary medicine, it has been investigated as a sole agent or in combination with metronomic chemotherapy in dogs with hemangiosarcoma, mammary carcinoma, inflammatory mammary carcinoma, and pulmonary carcinoma. Some possible benefits and promising results were found in lung and mammary carcinomas [77,78,79,80,81,82]. Thalidomide is an appealing drug in veterinary oncology, mainly due to its high safety profile, especially when combined with other chemotherapy drugs. Side effects reported from thalidomide are rare and mainly represented by mild sedation, but unexpected adverse events when combined with other chemotherapy agents have not been reported [77,80,81,82,83]. Thalidomide in combination with anti-VEGFR (vascular endothelial growth factor receptor) treatment such as toceranib phosphate and COX-2 inhibitors could increase the anti-angiogenic effect and improve the clinical efficacy in various types of cancer [77,80,81,83,84,85] (Table 1).

## 4. Possible Repurposed Drugs in Veterinary Oncology Not Yet Clinically Investigated in Dogs and Cats

### 4.1. Amlodipine

Several calcium channel blockers were proposed as anti-cancer molecules, including dihydropyridines such as amlodipine, nifedipine, nicardipine, and felodipine, but also non-dihydropyridines, including diltiazem and verapamil [86,87]. Amlodipine besylate is used to treat hypertension and supraventricular arrythmias, it has a very safe profile, and it is recommended as a first-line therapy for the management of hypertension in cats (except those with hyperthyroidism) [86,88]. Intracellular calcium is essential for the proliferation of various cell types [87], which can be partially explained by the Wnt signaling pathway, which includes two major intracellular transducers: intracellular free calcium and β-catenin (Figure 2). β-catenin is a transcription factor activator, but its molecule size of 92 KDa limits its diffusion through the nuclear membrane [89]. Nevertheless, the cytoplasmatic composition, such as the concentration of intracellular calcium, increases the nuclear membrane permeability [89,90]. Within the nucleus, β-catenin interacts with TCF/LEf, facilitating the transcription activation of several genes, including relevant proto-oncogenes such as c-Myc, c-Jun, and FRA1. This mechanism was demonstrated in an *in vitro* study with 58 histological samples of feline oral squamous cell carcinoma [90]. A quantitative immunohistochemical analysis revealed that the expression of c-Myc, CD1 (cyclin-D1), and FR1 was increased 2.3–3-fold compared to normal controls (*p* < 0.0001). Moreover, using a multilabel quantitative fluorophore technique, co-localization of the proteins and β-catenin in the nucleus was demonstrated [90]. The use of calcium channel blockers such as amlodipine is potentially promising for repurposing in veterinary oncology. Amlodipine also induced caspase-3/7 activation and downregulation of the anti-apoptotic protein Bcl2 in MDA-MB-231 breast cancer cells, also decreasing integrin β1 expression and impairing the invasive abilities of such cells *in vitro* [91]. In mice with xenografted human epidermoid carcinoma A431 cells, intraperitoneal administration of amlodipine (10 mg/kg) for 20 days resulted in retarded tumor growth and increased survival [87]. Amlodipine also attenuated tumor growth, when used in combination with gefitinib, in a study with non-small-cell lung cancer A549 xenografts [86]. Amlodipine could be used in combination with metronomic chemotherapy or conventional chemotherapy in various tumors, especially slow-growing carcinoma. However, in normotensive patients, blood pressure should be monitored during amlodipine treatment.

### 4.2. Amiloride

Amiloride is a potassium-sparing diuretic that acts through the inhibition of sodium channels on the nephrons distal convoluted tubules, resulting in the loss of sodium and water without potassium depletion [92]. It is also the only FDA-approved inhibitor of ATP-independent sodium–hydrogen exchangers [93]. While commonly used for the management of hypertension and congestive heart failure, it is also a promising drug for repurposing in oncology [92,93]. Sodium–hydrogen exchangers are ATP-independent antiporters that contribute to the maintenance of an intracellular near-neutral pH, even in an acidic extracellular environment [94]. Cancer cells usually have a high metabolic demand and promote aerobic glycolysis (Warburg effect), which results in a hypoxic and acidic microenvironment. Sodium–hydrogen exchangers are usually upregulated in cancer cells by hypoxia-inducible factors (HIF) and may contribute to cell survival and chemoresistance [93]. Therefore, targeting proton pumps such as sodium–hydrogen exchangers with amiloride may reduce the intracellular acidosis [93,94]. Amiloride may also induce the dephosphorylation (inactivation) of Akt through the inhibition of PI3 K, resulting in cytotoxicity mediated by TRAIL (tumor necrosis factor-related apoptosis-ligand), which may be related to the increased expression of p53 within the cytosol, the upregulation of the pro-apoptotic protein Bax, and the downregulation of the anti-apoptotic protein Bcl-xL, as demonstrated in canine osteosarcoma cell lines [93,94]. Amiloride inhibition of PI3 K/Akt potentiated the cytotoxicity mediated by EGFR tyrosine kinase inhibitors such as erlotinib in human pancreatic cancer cells [92]. Activating mutations in PI3K have been identified in canine osteosarcoma, and, given its aggressiveness and high metastatic rates, amiloride may be a useful drug to consider for the management of this cancer type [93,94].

### 4.3. Propranolol

Propranolol is a non-selective beta-adrenergic blocker used to treat various cardiovascular diseases [95,96]. In a preclinical model, propranolol has been found to regulate the cancer microenvironment and angiogenesis and inhibit metastatic disease [97,98,99]. Beta-adrenergic receptors are widely expressed in various cancers [100]. The activation of β-adrenergic receptors stimulates cyclic AMP (cAMP) synthesis, the phosphorylation of protein kinase A, and the activation of transcription factors, with consequent tumor cell proliferation, extracellular matrix invasion, angiogenesis, and matrix metalloprotease activation [101,102]. β-adrenergic activation also induces the expression of pro-inflammatory cytokines such as IL-6 and IL-8 in cancer and immune cells in the tumor microenvironment, which contributes to tumor growth [97,99,103] (Figure 3). Propranolol also stimulates apoptosis in cancer cells in vitro via the reduction of the levels of expression of NF-κB, VEGF, COX-2, MMP-2, and MMP-9 [104]. The anti-angiogenetic effect of propranolol seems mainly due to the downregulation of matrix metalloproteinase, MMP-2 and MMP-9, vascular endothelial growth factor (VEGF), and hypoxia inducible factor (HIF-1) signaling [105,106,107,108].

Propranolol treatment can overcome resistance to doxorubicin in leukemia cell lines and act in synergism with paclitaxel, vincristine, and 5-FU, both *in vitro* and *in vivo* in murine models [109,110,111]. Synergistic effects of propranolol have been also documented *in vitro* and *in vivo* with the tyrosine kinase inhibitors sunitinib and vemurafenib [109,112]

Propranolol has some *in vitro* immunomodulating anti-cancer effects and it can reverse the epinephrine inhibition of TNF-alpha-mediated cytotoxic T lymphocyte generation [113]. Propranolol can potentially also reduce the number and the immunosuppressive effect of regulatory T-cells [114]

It is known that propranolol is a phosphatidic acid phosphohydrolase (PAP) inhibitor [115]. PAP is an enzyme that converts phosphatidic acid (PA) to diacylglycerol, the second messenger activating the protein kinase C, which is involved in tumorigenesis and possible cancer progression, invasion, and metastasis [116]

The possibility of the anti-cancer effect of propranolol has been suggested by various epidemiological studies. In a systematic review and meta-analysis of observational studies of humans receiving beta blockers and suffering from various types of cancer, the use of a beta-blocker was associated with a significant reduction in the risk of death from breast cancer [117]. 

The potential stress-inhibitory pathway and immune-modulating effect of propranolol has been investigated in melanoma. In a retrospective study of human patients with melanoma, those receiving treatment with propranolol for concurrent diseases showed a reduced risk of melanoma progression [118].

The same group, more recently, in a small prospective study, found that the off-label administration of propranolol decreased the risk of recurrence by 80% for thick-type (>4 mm) melanoma [119]. 

Some clinical trials have investigated propranolol in combination with various other anti-cancer drugs in people suffering from advanced melanoma and pancreatic and breast carcinoma, with some promising results [120,121,122]. 

In one recent phase I study, propranolol was investigated in combination with anti-PD-1 checkpoint inhibitor pembrolizumab in advanced-stage melanoma [122]. The combination was safe and effective and no dose-limiting toxicity was noticed. Although this was only a phase I study with a small number of patients, 79% of patients achieved an objective response, a much higher response than expected with pembrolizumab alone (40–45%) [122,123].

Cancers arising from the endothelial vessels, benign angioma, and malignant angiosarcoma are rare in humans, and standard effective treatments have not been established [124]. Propranolol has shown significant efficacy in severe benign infantile angioma [125]. Beta-blockers, used alone or in combination with COX-2 inhibitors and metronomic chemotherapy, have shown some degree of efficacy in various sarcomas and in angiosarcoma in humans [5,6,126,127]. However, the efficacy of propranolol in angiosarcoma is based only on case reports and small case series rather than large studies [124].

Although clinical data on anti-cancer activity have not yet been published, some studies *in vitro* show promising results, and clinical trials investigating propranolol in combination with conventional chemotherapy for hemangiosarcoma in dogs are ongoing. 

In one study, canine osteosarcoma cell lines were treated with propranolol and carvedilol. The findings of the study showed that a sustained treatment with these compounds would reduce tumor cell growth and sensitize cancer cells to radiotherapy [128]. Propranolol has been found to sensitize canine hemangiosarcoma cells to chemotherapy also via Beta-AR-independent mechanisms. In one study, it was found that one of the mechanisms of doxorubicin drug resistance was caused by the accumulation of this drug in the intracellular lysosome, with consequent chemical inactivation [129]. Propranolol was shown to sensitize canine hemangiosarcoma cells to doxorubicin by increasing the concentration of doxorubicin inside the cancer cell cytoplasm and by decreasing the lysosomal accumulation and extracellular efflux [130].

Hemangiosarcoma is a common type of cancer affecting dogs [131]. As with human patients, visceral hemangiosarcoma is an aggressive disease with a very poor prognosis and limited treatment options. Effective adjuvant systemic treatments for this disease are lacking [132,133,134].

While aggressive and rapidly growing tumors such as hemangiosarcoma in dogs are unlikely to be effectively treated with propranolol, the combination of this drug with various chemotherapies, including metronomic or conventional chemotherapy, seems more rational and potentially promising [6,126].

The combination of propranolol with metronomic chemotherapy, similarly to that in humans, could be particularly promising in hemangiosarcoma treatment. The safety profile and low cost of both treatments are also appealing in veterinary medicine and could be potentially more beneficial than metronomic treatment alone [6,135]. The anti-angiogenic mechanism of propranolol could be synergistic with multi-kinase inhibitor toceranib and could be also investigated in various other solid tumors for which there are no effective systemic treatments, including advanced unresectable or metastatic soft tissue sarcomas and carcinoma [109]. Due to the immunomodulatory effects of propranolol and its possible synergistic effects with immunotherapy for melanoma in humans [122], a combination of propranolol with various immunotherapy strategies, including melanoma vaccines, could be trialed in the adjuvant treatment of oral melanoma in dogs.

### 4.4. Statins

Statins are a group of drugs that are used to lower cholesterol in humans. The main mechanism of action is to inhibit 3-hydroxy-3- methylglutaryl-coenzyme A (HMG-CoA) reductase, the enzyme involved in the synthesis of cholesterol [136]. Statins inhibit the conversion of HMG-CoA into mevalonate, which inhibits cholesterol formation. The inhibition of mevalonate also blocks the synthesis of isoprenoids [137]. Isoprenoids are a group of compounds that are important in the posttranslational modification (prenylation) of many intracellular signaling proteins and the GTPase cellular transduction proteins, such as RAS [138]. Inhibition of prenylation can alter membrane localization and so the activity of GTPase proteins such as RAS, which are often dysregulated in many human cancers [139].

In vitro statins seem to arrest cell cycle via the activation of Chk1 kinase and inhibition of cyclin A and CDK2 expression [140]. Statins inhibit phosphorylation and the activation of the Akt signaling pathway, reducing the proliferation, migration, and invasion of prostate cancer cells in vitro and in tumor xenografts [141]. Statins can induce oxidative stress and cancer cell death by inhibiting the formation of the electron transport chain intermediates such as CoQ10 that are involved in free radical scavenging [142]. 

Another possible mechanism behind the anti-cancer effect of statins is the depletion of cellular membrane cholesterol. It is known that some cancers cells have an increased amount of cholesterol and cholesterol rafts in their membranes compared to their normal counterparts. Decreased membrane cholesterol could impact the localization and activity of membrane growth factor receptor and disruption of the signaling cascade [143,144]. For example, cellular membrane cholesterol depletion can cause EGFR delocalization or detachment from membrane rafts and the subsequent alteration of the downstream signaling cascade [143,145,146].

Various studies in vitro have found anti-cancer efficacy for various statins, especially lipophilic statins such as simvastatin and atorvastatin. Lipophilic statins penetrate the cell membrane and enter the cell through passive diffusion and show higher pro-apoptotic activity compared to hydrophilic statins [147,148,149].

Hydrophobic statins were found to inhibit autophagy in vitro in a mevalonate pathway-dependent and -independent manner in rhabdomyosarcoma, glioma, lung adenocarcinoma, gastrointestinal carcinoma, breast carcinoma, and embryonic kidney cells [147,150,151,152,153,154,155]. Autophagy is a complex cellular phenomenon that involves lysosome-mediated cell digestion and the degradation of damaged, old, or normal proteins and organelles in response to stress or starvation or an increased growth demand. Autophagy seems to have a dual role in cancer, most likely having a protective role in cancer survival at earlier stages and a tumor-promoting effect in the late stage of cancer [156].

Lipophilic statins can also interact with the tumor microenvironment in various ways. In one study, simvastatin reduced lactic acid production and cancer sensitivity to monocarboxylate transporter 1, inhibiting HNSCC tumor growth [157]. In another study, simvastatin was able to switch tumor-associated macrophages (TAM) from an M2 to M1 phenotype, remodeling the tumor microenvironment and inhibiting epithelial–mesenchymal transition (EMT) [158].

Despite the numerous studies in vitro and in mouse models, the efficacy and benefit of statins for cancer patients are still controversial. Various small clinical trials have been performed, with contrasting results. A recent meta-analysis of randomized controlled trials investigating statin therapy in human cancer patients was conducted, evaluating ten studies, with a total of 1881 individuals with various cancers. The conclusion of the meta-analysis showed that in patients with advanced cancer and a prognosis <2 years, the addition of statins to standard anti-cancer therapy did not improve overall survival or progression-free survival [159]. However, in this study, various types of cancer with different stages were included, and specific subcategories (e.g., tumors with EGFR mutation versus wild type, etc.) were not considered when assessing survival in certain subgroups of patients. The failure of many clinical trials to prove a benefit for statins could be related to the poor clinical trial design, or poor selection of the tumor type, dose, and type of statin [160]. In a retrospective study of 19,974 human patients with lung cancer exposed to treatment with atorvastatin and simvastatin, a decrease in mortality risk and increase in survival time was found for squamous cell carcinoma patients, but not among those with small-cell lung cancer [161]. In a phase III randomized, double-blinded, placebo-controlled trial, 846 human patients with small-cell lung cancer were treated with standard-of-care chemotherapy with or without pravastatin. The study did not find any benefit of adding pravastatin to conventional chemotherapy for small-cell lung cancer [162]. In another, smaller randomized phase II clinical trial investigating gefitinib versus gefitinib plus simvastatin in patients with advanced non-small-cell lung cancer, the addition of simvastatin to gefitinib produced higher response rates and longer progression-free survival compared with gefitinib alone, in a subset of patients with wild-type EGFR non-adenocarcinomas. The authors concluded that simvastatin may improve the efficacy of gefitinib in this specific subgroup of gefitinib-resistant NSCLC patients [163].

Atorvastatin and fluvastatin were investigated recently in canine mammary tumor cells in vitro. Both statins showed cytotoxic effects in two mammary carcinoma cell lines (CMT9 and CMT47), with increased apoptosis and cell cycle arrest. Decreased yes-associated protein 1 (YAP) and transcriptional coactivator with PDZ-binding motif (TAZ) expression, which regulate transcriptional activity in various cancers, and reduced mRNA levels of key Hippo transcriptional target genes known to be involved in human breast cancer progression and chemoresistance were also found [164]. Atorvastatin has been already used in cats and dogs for various conditions, including hyperlipemia and heart failure, respectively. It has been reported to be safe in dogs and cats at the dose of 2 mg/kg and 5 mg/kg, respectively [165,166]. However, subclinical hemorrhages have been reported, albeit rarely, at doses higher than 20 mg/kg in healthy beagle dogs [167]. At standard dosages, atorvastatin could be an attractive drug to repurpose for cancer treatment in dogs and cats. Atorvastatin could be safely used in combination with metronomic/conventional chemotherapy or target therapy for slow-growing cancers such as some bladder carcinomas, nasal carcinomas, or others.

## 5. Conclusions

Repurposing drugs in small animal oncology is an attractive treatment strategy, and more studies should be performed to assess the safety and efficacy of various drugs for different types of tumors in dogs and cats. In the absence of large prospective clinical trials even in humans, it is still difficult to evaluate which drug would be the most beneficial in pets, and more studies need to be conducted in this field. However, the current evidence suggests that most repurposed drugs are unlikely to be significantly beneficial as single agents for rapidly growing cancers. It is more likely that repurposed drugs could be beneficial when used in combination with other drugs and/or with other chemotherapies or targeted therapies. The most difficult challenges in repurposing drugs in veterinary medicine will be the selection of the best compound for a particular type of cancer and establishing a safe and effective combination with other anti-cancer drugs. This review has not covered every drug/agent with potential for repurposing for cancer treatment, e.g., antiparasitic drugs, but has focused on those agents that the authors believe to hold the most interest or promise.

## Figures and Tables

**Figure 1 animals-13-00139-f001:**
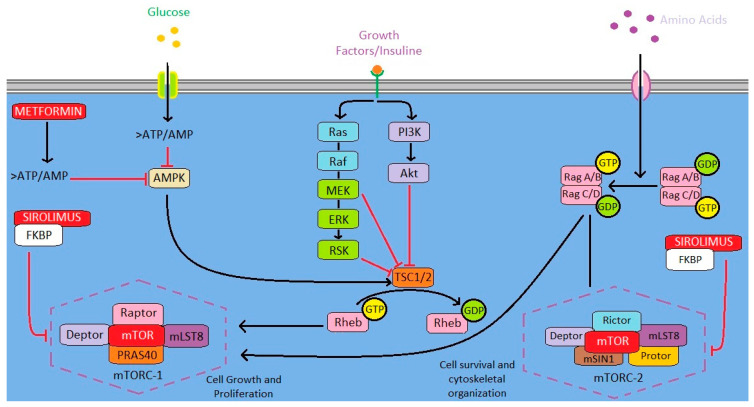
Metformin and sirolimus’ proposed anti-cancer effect’s mechanism of action.Metformin’s anti-cancer effect is most likely related to the inhibition of m-TOR through the AMPK-dependent pathway. Activated AMPK phosphorylates the tuberous sclerosis complex protein 2 (TSC2), which inhibits m-TOR complex 1 (m-TORC1), leading to cell growth arrest. Sirolimus binds to a family of intracellular binding proteins termed FKBPs (FK binding proteins) and such a complex acts as a specific inhibitor of m-TOR.

**Figure 2 animals-13-00139-f002:**
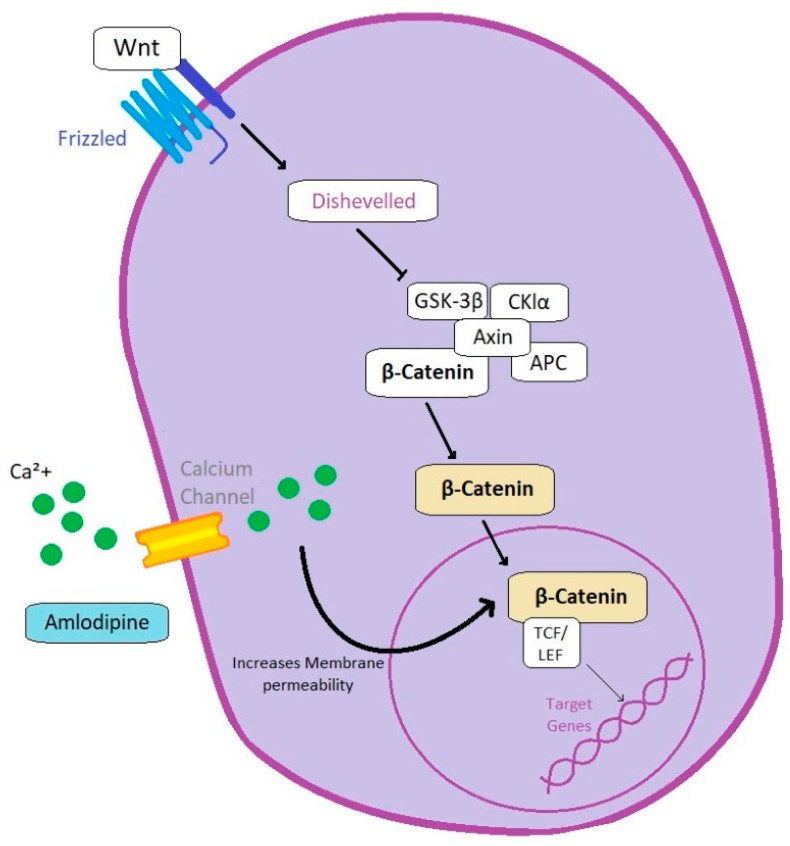
Amlodipine’s proposed anti-cancer effect. Amlodipine exerts its anti-cancer activity by blocking the cellular membrane calcium channels. The reduced intracellular calcium concentration reduces the nuclear membrane permeability to β-catenin, the nuclear translocation and activation of transcriptional factors, and the transcription of β-catenin target genes.

**Figure 3 animals-13-00139-f003:**
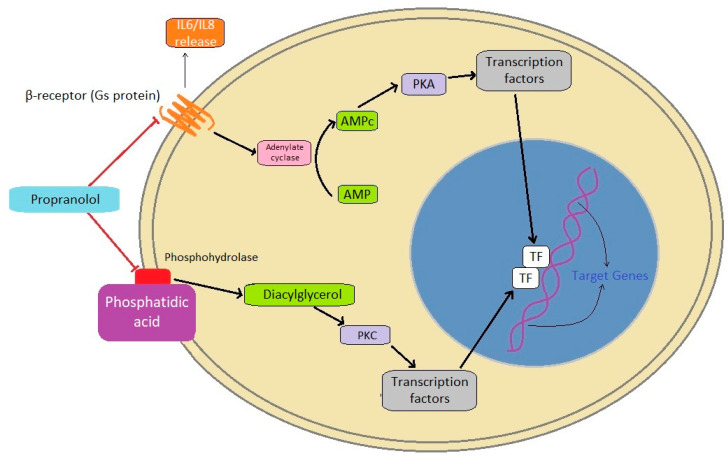
Proposed anti-cancer mechanism of action of propranolol. Activation of β-adrenergic receptors stimulates cyclic AMP (cAMP) synthesis, phosphorylation of protein kinase A, and activation of transcription factors, with consequent tumor cell proliferation, extracellular matrix invasion, angiogenesis, and matrix metalloprotease activation. β-adrenergic inactivation also reduces expression of pro-inflammatory cytokines such as IL-6 and IL-8 in cancer and immune cells in the tumor microenvironment, which impairs tumor growth. Propranolol is also a phosphatidic acid phosphohydrolase (PAP) inhibitor, blocking the synthesis of diacylglycerol, which activates the protein kinase C, involved in tumorigenesis and possible cancer progression, invasion, and metastasis.

**Table 1 animals-13-00139-t001:** Repurposed drugs with proposed mechanisms of action and results obtained in veterinary clinical trials.

Repurposed Drug	Action	Dose	Clinical Trial	Clinical Trial Results	Statistics	References
Auranofin	Inhibition of thioredoxin reductase with increased reactive oxygen species, oxidative stress, and cancer cell death	6 mg/dog < 15 kg PO q 3 days 9 mg/dog > 15 kg PO q 3 days	Dogs with osteosarcoma	Standard-care treatment (surgery + carboplatin) + auranofin (n = 40; OS = 329 days) *	*p* = 0.036	Endo-Munoz et al. (2019) [20]
				Historical control group with standard care (n = 26; OS = 240 days)		
Desmopressin	Anti-metastatic (inhibition of metastatic emboli formation and subsequent adherence of these emboli to target metastatic sites)	1 μg/kg IV 30 min preoperatively and 1 μg/kg IV 24 h postoperatively	Dogs with stage III or IV mammary carcinoma	Surgery + placebo (n = 10; DFI = 85 days; OS = 333 days)	*p* < 0.01 (DFI) and *p* = 0.05 (OS)	Hermo et al. (2008) [33]
				Surgery + desmopressin (n = 11; DFI = 608 days; OS > 600 days)		
		1 μg/kg IV 30 min preoperatively and 1 μg/kg IV 24 h postoperatively	Dogs with stage III or IV mammary carcinoma	Surgery + placebo (n = 10; DFI = 88 days; OS = 237 days)	*p* < 0.01	Hermo et al. (2011) [34]
				surgery + desmopressin (n = 18; DFI = 608 days; OS = 809 days)		
		3 mcg/kg SC preoperatively and 3 mcg/kg SC 24 h postoperatively	Dogs with mammary carcinoma	Surgery + placebo (n = 12; OS = 754 days)	*p* < 0.73	Sorenmo et al. (2020) [35]
				Surgery + desmopressin (n = 12; OS = 818 days)		
		1 μg/kg IV 30 min preoperatively and 1 μg/kg IV 24 h postoperatively	Cats with mammary carcinoma	Surgery (n = 45; DFI = 966 days)	*p* = 0.9	Wood et al. (2021) [32]
				Surgery + desmopressin (n = 15; DFI not reached)		
Doxycicline	Inhibition of MMP9 and NF-κB (anti-proliferative effect)	7.5 or 10 mg/kg PO q 12 h	Dogs with lymphoma	Doxycicline (n = 13; no objective response but one dog achieved stable disease for 6 weeks)	-	Hume et al. (2018) [44]
Losartan	Anti-metastatic (inhibition of CCL2-CCR2, monocyte recruitment, and tumor-associated macrophages)	10 mg/kg PO q 12 h.	Dogs with metastatic osteosarcoma	Toceranib + Losartan (n = 28; 50% of clinical benefit and 25% of objective response)	-	Regan et al. (2021) [52]
Metformin	Cell cycle arrest (AMPK activation leading to m-TOR inhibition and increased expression of p53)	10 mg/kg PO q 12 h.	Cats with various neoplasms (5 carcinomas, 2 cutaneous lymphomas and 2 injection site sarcomas)	9 cats Metformin (n = 9; 2 cats with skin SCC had modest measurable response)	-	Wypij (2015) [66]
Sirolimus (rapamycin)	Cell cycle arrest (m-TOR inhibition)	0.1 mg/kg on either a Monday through Friday schedule or Monday–Wednesday–Friday schedule for 4 consecutive weeks	Dogs with osteosarcoma	Standard-care treatment (surgery + carboplatin) (n = 157; OS = 282 days)	*p* > 0.05	LeBlanc et al. (2021) [73]
				Standard-care + sirolimus (n = 152; OS = 280 days)		
Thalidomide	Anti-angiogenic properties, decreased VEGF and TNF-alpha	20 mg/kg PO q 24 h for 3 months, followed by 10 mg/kg PO q 24 h.	Dogs with stage V mammary carcinoma	Surgery (n = 5; OS = 150 days)	*p* < 0.0001 (except between groups treated with surgery or surgery + maximum tolerated dose of chemotherapy)	Campos et al. (2018) [77]
				Surgery + maximum tolerated dose of chemotherapy (n = 3; OS = 148 days)		
				Surgery + maximum tolerated dose of chemotherapy + metronomic chemotherapy (n = 6; OS = 376.5 days)		
				Surgery + maximum tolerated dose of chemotherapy + thalidomide (n = 13; OS = 463 days)		
		2 mg/kg PO q 24 h	Dogs with inflammatory mammary carcinoma	Piroxicam + thalidomide + toceranib (n = 14; OS = 59 days)	*p* = 0.032	Rossi et al. (2018) [80]
				Radiotherapy + piroxicam + thalidomide + toceranib (n = 4; OS = 180 days)		
		8.7 mg/kg PO q 24 h	Dogs with stage II (n = 10) and III (n = 5) splenic haemangiosarcoma	Surgery and thalidomide (OS = 172 days)	-	Bray et al. (2018) [82]
		2 mg/kg PO q 24 h	Dogs with advanced primary lung carcinoma	Metronomic chemotherapy (low-dose cyclophosphamide + piroxicam + thalidomide) (n = 25; OS = 139 days)	*p* < 0.007 (difference between metronomic/thalidomide group and the three remaining)	Polton et al. (2018) [81]
				Surgery (n = 36; OS = 92 days)		
				Maximum tolerated dose of chemotherapy (n = 11; OS = 61 days)		
				No oncologic treatment (n = 19; OS = 60 days)		

* Improved outcome was attributable only to male dogs (472 days) with better OS than females (240 days) with *p* = 0.009. OS—overall survival, DFI—disease-free interval, SCC—squamous cell carcinoma.

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
