# Peer review of "Repurposing Drugs in Small Animal Oncology"

_animals, 2022, doi:10.3390/ani13010139_

Round 1
Reviewer 1 Report
Brief summary
Article entitled "Repurposing drugs in small animal oncology" is a review article describing the possibilities of using various commonly used drugs in the treatment of cancer in animals. The authors decided to describe a few drugs that they believe have the greatest potential: Auranofin, Desmopressin, Doxycycline, Losartan, Metformin, Sirolimus, Thalidomide, Amlodipine, Amiloride, Propranolol and Statins (atorvastatin).
Broad comments
The subject of the article “Repurposing drugs in small animal oncology” fits the aims and scope of Animals and is important and interesting to both clinicians and scientists. It is of great practical importance. The article raises an important topic, it is well written, which is why it is valuable for the veterinarian community.
The article is written in a logical, clear and linguistically correct way but there are minor formatting errors in the text. The article includes figures and tables that help understand the topic.
Specific commentst
Page: 2 Line: 54 - 57
Repeated and unfinished sentence - please correct
Why did the authors not mention the possibility of using some antiparasitic drugs in the treatment of cancer? I suggest at least adding information about ivermectin - so that the article covers all the most important issues.
Author Response
Brief summary
Article entitled "Repurposing drugs in small animal oncology" is a review article describing the possibilities of using various commonly used drugs in the treatment of cancer in animals. The authors decided to describe a few drugs that they believe have the greatest potential: Auranofin, Desmopressin, Doxycycline, Losartan, Metformin, Sirolimus, Thalidomide, Amlodipine, Amiloride, Propranolol and Statins (atorvastatin).
Broad comments
The subject of the article “Repurposing drugs in small animal oncology” fits the aims and scope of Animals and is important and interesting to both clinicians and scientists. It is of great practical importance. The article raises an important topic, it is well written, which is why it is valuable for the veterinarian community.
The article is written in a logical, clear and linguistically correct way but there are minor formatting errors in the text. The article includes figures and tables that help understand the topic.
Thank you for your comment we really appreciate your opinion.
Specific comments
Page: 2 Line: 54 - 57
Repeated and unfinished sentence - please correct
Thank you for your comment we have corrected this in the text.
Why did the authors not mention the possibility of using some antiparasitic drugs in the treatment of cancer? I suggest at least adding information about ivermectin - so that the article covers all the most important issues.
Thank you for your comment, we really appreciate your suggestion. The list of possible repurposing drugs for use in cancer is very extensive. We have as well considered the possibility to add antiparasitic drugs like ivermectin and Fenbendazole, however we decided to not add this drug as a promising drug, but that was just an author choice. We however specify in the conclusion that this is not a complete review of all the possible repurposing drugs and that we selected only the one that we believed could be more promising.
Reviewer 2 Report
Major comments:
- Many small studies are described but a clear summary of which drugs have support for repurposing via systematic reviews and meta-analyses is needed. In particular, whether any have been conducted for companion animals.
- No discussion of side effects or detriment arising from repurposing of any of the drugs is given.
- Structure:
o For each drug listed, a consistent structure would help the reader take in the content. The section on Losartan was a good example of a clear structure with Name + target + clinical indication + mechanism, all outlined in a clear order, prior to going into details of studies.
o Inconsistent level of detail into mechanisms per drug given. A little more consideration of which details are/aren’t supporting the aims of the manuscript is needed
o Overall structure for each section needs to be clearer e.g. introduce the drug, what is the evidence for (esp stystematic reviews/meta-analyses), what is the evidence against or limitations of the literature. E.g. statins section is very long and rambling with a large focus on in vitro data and comes across as speculative
- Paragraph starting at line 56 outlining advantages of repurposing rugs could be far more concise
- Conclusions: “It is the authors opinion” should be replaced with a conclusion based on the evidence reviewed
- Table 1:
o Formatting needs improving. Very cramped and difficult to read E.g. no spacing between digits and units, around “=” signs etc
o Referencing style is not consistent with Vancouver style of the rest of the paper. Correcting this would free up space to better format the rest of the table
o Separate columns for species, dose, route of admin, n’s per group would make for a clearer table
o Typo: “Objectctive”
- Figures:
o Not clear whether these are original or adapted from other sources
o Not clear how/why the three figures were selected to be included over any of the other drug mechanisms- they are full of mechanistic biomolecular detail which doesn’t seem the best use of the space. A figure(s) that summarises the paper as a whole, instead of select parts of it, may be of better assistance to the reader
o Figure 2 not referenced in text
o Text is too small to read in all figures
Minor comments:
- Many formatting inconsistencies and errors:
o Spacing between text and in-text citations
o Spacing between the digits of a measure and the units
o Spacing between words – instances of double space or no space after a full stop
o Formatting of paragraphs for some sections needs revising e.g. sections for propranolol and statins have many short paragraphs, some only one sentence long.
- Grammatical/syntactical errors or improvements needed:
o General:
§ It is often unclear whether the authors are referring to animals or humans. Greater care should be taken in making clear distinctions. E.g. using the term “patients” for humans and “subjects” for animals.
§ “humans” should be used instead of “people”
§ Latin terms e.g. in vitro, in vivo should be italicised
o Specific
§ Line 31: “using anti-cancer repurposing” could be “repurposing anti-cancer”
§ Line 39: us of word “Probably” suggests lack of conviction and expertise
§ Line 53: “only few”
§ Line 54: sentence is incomplete
§ Line 98: “ability to release of factor…”
§ Line 176: “may occur in a dose-dependent manner” the use of the term “may” indicates lack of comprehension of the evidence; “dose-dependent” should be “concentration-dependent” when referring to in vitro (cell-line) studies.
§ Line 177: “in in”
§ Line 242-243: sentence repeated from start of paragraph
§ Line 248: “tumor necrosis factor TNF” should have TNF in parentheses
§ Line 385: “B-blokers” should be “Beta” or define the abbreviation first
§ Line 421: is “atorvastatin” necessary since other statins are discussed?
§ Line 427: “like farnesyl pyrophosphate” wither not necessary or replace “e.g.” for “like”?
§ Line 428: Ras proteins could be better introduced
Author Response
Major comments:
- Many small studies are described but a clear summary of which drugs have support for repurposing via systematic reviews and meta-analyses is needed. In particular, whether any have been conducted for companion animals.
Thank you for your comment, we agreed that unfortunately there are not many systematic reviews on the real efficacy for most of these drugs. Systematic review assessing the efficacy of any of the repurposed drugs we discussed are lacking in dogs and cats and we made this clear in the conclusion. The few systematic review we found for each drug have been reported and specified in each paragraph in the text.
- No discussion of side effects or detriment arising from repurposing of any of the drugs is given.
Thank you for your comment. Repurposed drugs are old drugs that are already well known for their primary indication and side effects. We prefer only to focus on the potential anti-cancer activity and we specified adverse events and safety profile when the drugs were used as anticancer especially if a different dosage from the primary indication or in combination with other anti-cancer drugs. We specify this in the text.
- Structure:
o For each drug listed, a consistent structure would help the reader take in the content. The section on Losartan was a good example of a clear structure with Name + target + clinical indication + mechanism, all outlined in a clear order, prior to going into details of studies.
Thank you for your comments, we tried to follow already this structure. We modify the text to follow a clearer structure as you suggested when was not done already.
- Inconsistent level of detail into mechanisms per drug given. A little more consideration of which details are/aren’t supporting the aims of the manuscript is needed
Thank you for your comment. The level and details reported about the mechanism of action depend on the available literature, while some drugs have been extensively studied for their possible anticancer mechanisms, some are much less so. We tried to make more homogeneous all the paragraphs and remove excessive invitro details for some drugs. Modifications have been made in the text.
- Overall structure for each section needs to be clearer e.g. introduce the drug, what is the evidence for (esp stystematic reviews/meta-analyses), what is the evidence against or limitations of the literature. E.g. statins section is very long and rambling with a large focus on in vitrodata and comes across as speculative
Thank you for your comment. We tried to make this clearer and specify (when was not), the evidence and type of study available and reported. In the statins paragraph some details were removed.
- Paragraph starting at line 56 outlining advantages of repurposing rugs could be far more concise
Thank you for your comment, we think this is already very concise. It is very important to describe in detail all the advantages of drug repurposing in vet medicine.
- Conclusions: “It is the authors opinion” should be replaced with a conclusion based on the evidence reviewed
Thank you for your comment we changed this in the text
- Table 1:
o Formatting needs improving. Very cramped and difficult to read E.g. no spacing between digits and units, around “=” signs etc
o Referencing style is not consistent with Vancouver style of the rest of the paper. Correcting this would free up space to better format the rest of the table
o Separate columns for species, dose, route of admin, n’s per group would make for a clearer table
- Typo: “Objectctive”
Thank you for your comments, all the corrections were made in the text
- Figures:
o Not clear whether these are original or adapted from other sources
- Not clear how/why the three figures were selected to be included over any of the other drug mechanisms- they are full of mechanistic biomolecular detail which doesn’t seem the best use of the space. A figure(s) that summarises the paper as a whole, instead of select parts of it, may be of better assistance to the reader
Thank you for your comment. We decided to put some examples of mechanism of action of classic and very common drugs to illustrate as few examples, the potential anti-cancer mechanisms of action. We agree with you, but It is not possible to add a picture for every drug or a summative picture for all. We specify this in the description of the pictures.
o Figure 2 not referenced in text
o Text is too small to read in all figures
Minor comments:
- Many formatting inconsistencies and errors:
o Spacing between text and in-text citations
o Spacing between the digits of a measure and the units
- Spacing between words – instances of double space or no space after a full stop
Thank you for your comment we corrected all the formatting and grammar errors we found in the text
o Formatting of paragraphs for some sections needs revising e.g. sections for propranolol and statins have many short paragraphs, some only one sentence long.
- Grammatical/syntactical errors or improvements needed:
Thank you for your comment we corrected all the formatting and grammar errors we found in the text
o General:
- It is often unclear whether the authors are referring to animals or humans. Greater care should be taken in making clear distinctions. E.g. using the term “patients” for humans and “subjects” for animals.
Thank you for your comments, we changed this in the text. We prefer refer to canine/feline patient and human patients
- “humans” should be used instead of “people”
Thank you for your comment this has been changed in the text
- Latin terms e.g. in vitro, in vivoshould be italicized
Thank you for your comment this has been changed in the text
o Specific
- Line 31: “using anti-cancer repurposing” could be “repurposing anti-cancer”
Thank you for your comment this has been changed in the text
- Line 39: us of word “Probably” suggests lack of conviction and expertise
Thank you for your comment probably has been removed from the text
- Line 53: “only few”
Thank you for your comment this has been changed in the text
- Line 54: sentence is incomplete
Thank you for your comment this has been changed in the text
- Line 98: “ability to release of factor…”
Thank you for your comment this has been changed in the text
- Line 176: “mayoccur in a dose-dependent manner” the use of the term “may” indicates lack of comprehension of the evidence; “dose-dependent” should be “concentration-dependent” when referring to in vitro (cell-line) studies.
Thank you for your comment this has been changed in the text
- Line 177: “in in”
Thank you for your comment this has been changed in the text
- Line 242-243: sentence repeated from start of paragraph
Thank you for your comment this has been removed from the text
- Line 248: “tumor necrosis factor TNF” should have TNF in parentheses
Thank you for your comment this has been changed in the text
- Line 385: “B-blokers” should be “Beta” or define the abbreviation first
Thank you for your comment this has been changed in the text
- Line 421: is “atorvastatin” necessary since other statins are discussed?
Thank you for your comment this has been removed from the text
- Line 427: “like farnesyl pyrophosphate” wither not necessary or replace “e.g.” for “like”?
Thank you for your comment this has been changed in the text
- Line 428: Ras proteins could be better introduced
Thank you for your comment this has been changed in the text
